# Learning-In-The-Loop Optimization: End-To-End Control And Co-Design of Soft Robots Through Learned Deep Latent Representations

**Andrew Spielberg, Allan Zhao, Tao Du, Yuanming Hu, Daniela Rus, Wojciech Matusik**
CSAIL
Massachusetts Institute of Technology
Cambridge, MA 02139
aespielberg@csail.mit.edu, azhao@mit.edu, taodu@csail.mit.edu
yuanming@mit.edu, rus@csail.mit.edu, wojciech@csail.mit.edu

## Abstract

Soft robots have continuum solid bodies that can deform in an infinite number of ways. Controlling soft robots is very challenging as there are no closed form solutions. We present a learning-in-the-loop co-optimization algorithm in which a latent state representation is learned as the robot figures out how to solve the task. Our solution marries hybrid particle-grid-based simulation with deep, variational convolutional autoencoder architectures that can capture salient features of robot dynamics with high efficacy. We demonstrate our dynamics-aware feature learning algorithm on both 2D and 3D soft robots, and show that it is more robust and faster converging than the dynamics-oblivious baseline. We validate the behavior of our algorithm with visualizations of the learned representation.

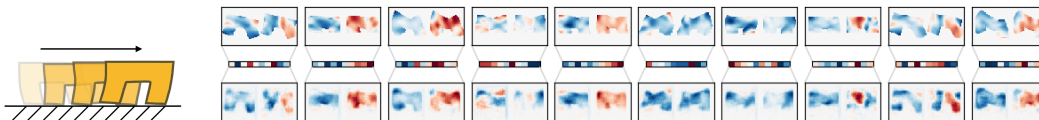

Figure 1: Our algorithm learns a latent representation of robot state which it uses as input for control. Above are velocity field snapshots of a soft 2D biped walker moving to the right (top), the corresponding latent representations (middle), and their reconstructions (bottom) from our algorithm. In each box, the $x$ (left) and $y$ (right) components of the velocity fields are shown; red indicates negative values, blue positive.

## 1 Introduction

Recent breakthroughs have demonstrated capable computational methods for both controlling (Heess et al. [2017], Schulman et al. [2017], Lillicrap et al. [2015]) and designing (Ha et al. [2017], Spielberg et al., Wampler and Popović [2009]) rigid robots. However, control and design of *soft* robots have been explored comparatively little due to the incredible computational complexity they present. Due to their continuum solid bodies, soft robots' state dimensionality is inherently infinite. High, but finite dimensional approximations such as finite elements can provide robust and accurate forward simulations; however, such representations have thousands or millions of degrees of freedom, making them ill-suited for most control tasks. To date, few compact, closed-form models exist for describing soft robot state, and none apply to the general case. In this paper, we address the problem of learning low-dimensional robot state while simultaneously optimizing robot control and/or material parameters. In particular, we require a representation applicable to physical control of real-world robots.

We propose a computer vision-inspired approach which makes use of the robot's observed dynamics in learning a compact observation model for soft robots. Our task-centric method interleaves

controller (and material) optimization with learning low-dimensional state representations. Our "learning-in-the-loop optimization" method is inspired by recent advances in hybrid particle-grid-based differentiable simulation techniques and deep, unsupervised learning techniques. In the *learning phase*, simulation grid data is fed into a deep, variational convolutional autoencoder to learn a compact latent state representation of the soft robot's motion. In the *optimization phase*, the learned encoder function creates a compact state description to feed into a parametric controller; the resulting, fully differentiable representation allows for backpropagating through an entire simulation and directly optimizing a simulation loss with respect to controller and material parameters.

Because learning is interleaved with optimization, learned representations are catered to the task, robot design (including, *e.g.*, discrete actuator placement), and environment at hand, and not just the static geometry of the soft structure. Because of our judicious choice of a physics engine which operates (in part) on a grid, we are able to easily employ modern, deep learning architectures (convolutional neural networks) to extract robust low-dimensional state representations while providing a representation amenable to real-world control through optical flow. Because of our fully-differentiable representation of the controller, observations, and physics, we can directly co-design robot performance.

To our knowledge, our pipeline is the first end-to-end method for optimizing soft robots without the use of a pre-chosen, fixed representation, minimizing human overhead. In this paper, we contribute: *1)* An algorithm for control and co-design of soft robots without the need for manual feature engineering; *2)* experiments on five model robots evaluating our system's performance compared to baseline methods; *3)* visualizations of the learned representations, validating the efficacy of our learning procedure.

## 2  Related Work

**Dimensionality Reduction for Control** A compact, descriptive latent space is crucial for tractably modeling and controlling soft robots. Methods for extracting and employing such spaces for control typically fall into two categories: *a)* analytical methods, and *b)* learning-based methods.

Analytical methods examine the underlying physics and geometry of soft structures in order to extract an optimal subspace for capturing low-energy (likely) deformations. Most popular among these methods are modal bases [Sifakis and Barbic, 2012], formed by solving a generalized eigenvalue problem based on the harmonic dynamics of a system. These methods suffer from inadequately modeling actuation, contact, and tasks and only represent a linear approximation of system's dynamics. Still, such representations have been successfully applied to real-time linear control (LQR) in Barbič and Popović [2008] and Thieffry et al. [2018], and (with some human labeling) animation [Barbič et al., 2009], but lack the physical accuracy needed for physical fabrication. In another line of work, Chen et al. [2017] presented a method for using analytical modal bases in order to reduce the degrees of freedom of a finite element system for faster simulation while maintaining physical accuracy. However, the resulting number of degrees of freedom are still impractical for most modern control algorithms. For the specific case of soft robot arms, geometrically-inspired reduced coordinates may be employed. Della Santina et al. [2018] developed a model for accurately and compactly describing the state of soft robot arms by exploiting segment-wise constant curvature of arms.

Learning-based methods, by contrast, use captured data in order to learn representative latent spaces for control. Since these representations are derived from robot simulations or real-world data, they can naturally handle contact, actuation, and be catered to the task. Goury and Duriez [2018] demonstrated some theoretical guarantees on how first-order model reduction techniques could be applied to motion planning and control for real soft robots. As drawbacks, their method is catered to FEM simulation, requires *a priori* knowledge of how the robot will move, and representations are never re-computed, making it ill-suited to co-design where dynamics can change throughout optimization.

Two works from different domains have similarities to our work. Ma et al. [2018] applied deep-learning of convolutional autoencoders in the context of controlling rigid bodies with directed fluids. Our algorithm shares high-level similarities, but operates in the domain of soft robot co-optimization and exploits simulation differentiability for fast convergence. Amini et al. [2018] employed latent representations for autonomous vehicle control in the context of supervised learning on images.

**Co-Design of Soft Robots** There exist two main threads of work in which robots are co-designed over morphology and control — gradient-free and gradient-based. Most of the work in model-free co-optimization of soft-robots is based on evolutionary algorithms. Cheney et al. [2013], Corucci

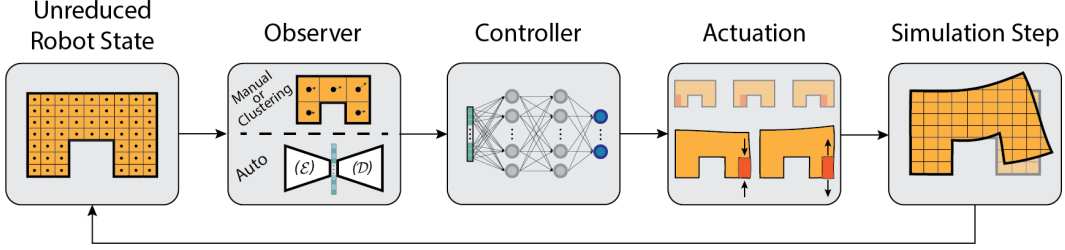

| Unreduced Robot State | Observer | Controller | Actuation | Simulation Step |

Figure 2: At each step of our simulation, the following procedure is performed. First, the unreduced state is fed into an observer function — centroids of a segmentation, as in Hu et al. [2019], or, as we demonstrate in this paper, an automatically learned latent space. Regardless, the observer outputs features to be processed by a parametric controller, which converts these features to actuation signals. Finally, the actuation is fed into our MPM simulator, which performs a simulation step. The entire pipeline is differentiable and therefore we can compute derivatives with respect to design variables even when executing the workflow for many steps.

et al. [2016], and Cheney et al. [2018] have demonstrated task-specific co-optimization of soft robots over materials, actuators, and topology. These approaches are less susceptible to local minima than gradient-based approaches but are vastly more sample inefficient. For instance, a single evolved robot in Cheney et al. [2013] requires 30000 forward simulations; by comparison, optimized robots in our work are optimized in the equivalent of 400 simulations (treating gradient calculation as equal to 3 forward simulations). Further, their approach was limited to simple open-loop controllers tied to form, while ours solves for more robust, closed-loop control.

While some algorithms exist for gradient-based co-optimization of rigid robots (Wampler and Popović [2009], Spielberg et al., Ha et al. [2017], Wang et al. [2019]), results in model-based co-optimization of soft robots have been sparse. Closest to our work, Hu et al. [2019] presented a method for gradient-based co-optimization of soft robotic arms using a fully-differentiable simulator based on the material point method (MPM). As a limitation, their method relied on ad-hoc features that needed to be labeled at the time the robot topology was specified and could not be easily measured in the physical world, making them ill-suited for physical control tasks. Our work addresses this shortcoming.

## 3   Overview and Preliminaries

We seek an algorithm for co-optimizing soft robots over control and design parameters without manually prescribing a state description for the controller to observe. Our solution will be to periodically learn an updated observation model from the simulation data generated during optimization. For the remainder of this paper, we refer to the dimensionally reduced representation of the soft robot as the *latent* representation and the unreduced representation as the *full* representation. To avoid confusion, we use the term *learning* to refer to the procedure of learning a latent representation and the term *optimization* to exclusively refer to the procedure of improving a robot's controller or design.

A full overview of our system is shown in Fig. 2. At each time step, the full representation is fed into a (learned) *observer* function, which reduces it down to a latent representation. The latent representation is fed into an (optimized) controller function, which produces control signals for the robot's actuators. Those control signals are applied to the full robot state to simulate the robot forward one time step, producing the next full state. At the end of the simulation, a specified *final loss* function $L$ is computed. Direct optimization of this loss function over controller and physical design parameters is possible since each component of our system, including our simulator, is differentiable.

Formally, let $\boldsymbol{v}^t \in \mathbb{R}^u$ denote a robot's full state at time $t$, let $\mathbf{q}^t \in \mathbb{R}^r$ denote the corresponding latent space, and let $\mathbf{u}^t \in \mathbb{R}^m$ denote the actuation control signal at time $t$. The observer function, $\mathcal{O} : \mathbb{R}^u \to \mathbb{R}^r$ maps a full state to a latent state and is governed by observer parameters $\boldsymbol{\Theta}$. The controller function, $\mathcal{C} : \mathbb{R}^r \to \mathbb{R}^m$ maps a reduced state to deterministic actuation output and is governed by control parameters $\boldsymbol{\theta}$. The simulation step function, $f : \mathbb{R}^u \times \mathbb{R}^m \to \mathbb{R}^u$, time steps the system for some specified $\Delta t$ given the full state and the actuation, and is governed by physical design parameters $\boldsymbol{\phi}$. In other words, $\boldsymbol{v}^{t+\Delta t} = f(\boldsymbol{v}^t, \mathcal{C}(\mathcal{O}(\boldsymbol{v}^t; \boldsymbol{\Theta}); \boldsymbol{\theta}); \boldsymbol{\phi})$. For brevity, we will omit writing the parameterizations explicitly except when necessary. The final loss $L : \mathbb{R}^u \to \mathbb{R}$ at final time $T$ operates on some final full state $\boldsymbol{v}^T$ to produce a scalar loss; this could be the distance the robot has traveled, its final velocity, *etc.* - anything that's dependent on the robot's final state. Computing $L$ amounts to iteratively applying $f$ to generate states $\boldsymbol{v}^0, \boldsymbol{v}^{\Delta t}, \boldsymbol{v}^{2\Delta t} \ldots \boldsymbol{v}^T$ and applying $L$ to the final

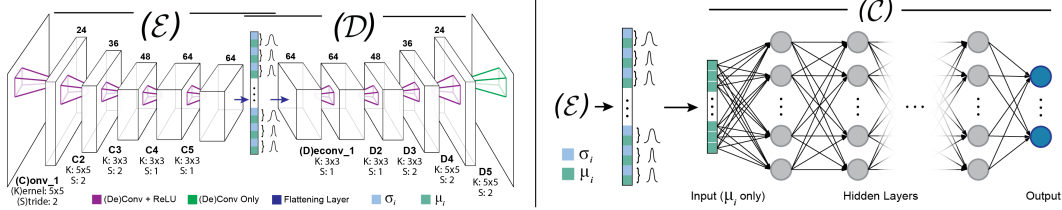

Figure 3: **Left:** The architecture of our convolutional variational autoencoder. The autoencoder takes in 2-channel pixel grid data as input, with each channel representing the $x$ or $y$ velocity field at that pixel. We use five layers of strided convolutions followed by ReLU operations. This is followed by a final flattening operation which coalesces the weights into latent variables. The latent variables parameterize Gaussians, used in our variational loss ($\mathcal{L}_v$). The architecture is mirrored on the opposite side (without a final ReLU, to allow for negative outputs). The 3D version is completely analogous, but takes in 3-channel voxel grid velocity field data and applies 3D convolutions. Above, the filter sizes are specified with $K$, and the strides are specified with $S$. **Right:** At inference time, simulation data is fed into the encoder, $\mathcal{E}$, which produces a latent $[\boldsymbol{\mu}, \boldsymbol{\sigma}]^\dagger$ vector. The mean $\boldsymbol{\mu}$ variables are then fed as inputs to the controller, $\mathcal{C}$.

one. We use $\mathcal{S}$ to denote the full process of simulating a robot and then computing the loss value $l$; in other words, $l = L(\boldsymbol{v}^T) = \mathcal{S}(L, \boldsymbol{v}^0)$ for some simulation time length $T$. The chain rule can then be used to backward-propagate through this chain of functions to compute gradients for optimization. In our algorithm, the learning phase learns $\boldsymbol{\Theta}$ while the optimization step optimizes $\phi$, and $\theta$.

$$
\begin{aligned}
\underset{\boldsymbol{\theta}, \boldsymbol{\phi}}{\text{minimize}} \quad & L(\boldsymbol{v}^T) \\
\text{where} \quad & \forall t, \ \boldsymbol{v}^{t+\Delta t} = f(\boldsymbol{v}^t, \mathcal{C}(\mathcal{O}(\boldsymbol{v}^t; \boldsymbol{\Theta}); \boldsymbol{\theta}); \boldsymbol{\phi}) \\
\text{subject to} \quad & \boldsymbol{\phi}_{\min} \leq \boldsymbol{\phi} \leq \boldsymbol{\phi}_{\max}
\end{aligned}
$$

Though $\boldsymbol{\Theta}$ is not part of the optimization, it is an auxiliary variable that must be learned in tandem.

## 4 Method

**Simulation and Data Generation** We use a simulator based on ChainQueen [Hu et al., 2019], the differentiable Moving Least Squares Material Point Method (MLS-MPM) [Hu et al., 2018] simulator, for the underlying physics of robots. In ChainQueen, robots are represented as collections of particles, and a background velocity grid is used for particle interaction and is exposed to users. ChainQueen also provides *analytical gradients* of functions of simulation trajectories with respect to controller parameters and robot design. Our algorithm is not simulator specific, and can operate on any fully differentiable simulator where differentiable grid velocity data can be extracted.

In the remaining of this section we describe our LITL optimization algorithm. First, we assume we have a large dataset of robot motion data from the simulator, representative of the way the robot will move when completing the prescribed task, and describe the learning phase of the algorithm. Next, we describe how we use the simulation data to optimize the controller and design. Finally, we describe how to combine these two phases into a cohesive, complete algorithm.

### 4.1 Learning

During the learning phase, we seek to learn a compact, expressive representation of robot state to feed to the controller. As input, learning takes in snapshots of robot simulation; namely, the robot's velocity field on a background grid. Note that this field implicitly also provides robot positional information. As output, weights for an observer function with a descriptive latent space are learned.

In particular, we learn a variational autoencoder [Kingma and Welling, 2013, Rezende et al., 2014] that takes, as input, a state description of a robot and minimizes the reconstruction cost of said state. Our assumption is that features which allow reconstruction are highly expressive. Fig. 3 presents the architecture we used for all experiments. We experimented with various network depths; ours was chosen for stability and generality across experiments. We use a convolutional architecture, which operates naturally on input velocity grid data and generalizes to robot translation due to equivariance.

Formally, for an unreduced $u$-dimensional object, let $\mathcal{E}: \mathbb{R}^u \to \mathbb{R}^r$ be an encoder function with parameter weights $\boldsymbol{\Theta}_\mathcal{E}$, and $\mathcal{D}: \mathbb{R}^r \to \mathbb{R}^u$ be a decoder function with parameter weights $\boldsymbol{\Theta}_\mathcal{D}$. For an

input training dataset grid velocity data $\Upsilon$, the reconstruction loss is defined as:

$$\mathcal{L}_R(\Upsilon) = \frac{1}{|\Upsilon|} \sum_{\boldsymbol{v} \in \Upsilon} \|\mathcal{D}_{\Theta_{\mathcal{D}}}(\mathcal{E}_{\Theta_{\mathcal{E}}}(\boldsymbol{v})) - \boldsymbol{v}\|_2^2.$$

We omit the details of the VAE formulation, which adds a variational representation and regularization term. For a more extensive treatment, please refer to [Doersch, 2016]. We minimize this loss by performing mini-batch stochastic gradient descent on our input dataset. For updates, we employ the Adam [Kingma and Ba, 2014] first-order optimizer. We also experimented with a non-variational autoencoder. However, in the majority of experiments, that network overfit to a $1D$ manifold. This caused control optimization to fail, quickly resulting in unpredictable, erratic behaviors. The regularization from the variational formulation is necessary for avoiding collapse in the latent space.

## 4.2    Optimization

Our optimization procedure is similar to that of Hu et al. [2019]. At each optimization iteration, we compute $\nabla_{\boldsymbol{\theta},\boldsymbol{\phi}} L$, providing a gradient of our loss with respect to all of our decision variables. We then use this gradient to apply a gradient descent update step to our parameters. Finally, we account for potential bounds in our design variables (*e.g.*, maximum and minimum Young's Modulus) by projecting the variable bounds back to the feasible region; *i.e.* $\phi_i \leftarrow \max(\min(\phi_i, \phi_{\max}), \phi_{\min})$. In practice, we use the Adam optimizer. At each iteration of the optimization, during forward simulation, we record snapshots of the grid data to be used in the learning phase.

## 4.3    Algorithm

Our algorithm is an *alternating minimization*. First, we optimize the robot controller and design parameters for a fixed number of iterations, during which we record snapshots of grid velocities. Then, we use these grid velocities to learn an observer for a fixed number of iterations. With the observer and latent representation improved, we return to our optimization procedure, and keep alternating until convergence. The initial grid velocity dataset is generated from simulating just once with the initial, untrained controller. This is enough to bootstrap our learning. Two key design decisions are discussed below:

**Alternative vs. Simultaneous Minimization** Learning a descriptive latent encoding is harder than optimizing the controller, and therefore requires more minimization iterations. When trained simultaneously, the controller tends to get trapped into a local minimum under a non-descriptive latent space. Performance-wise, evaluating $\nabla_{\boldsymbol{\Theta}} L$ is orders of magnitude more expensive than evaluating $\nabla_{\boldsymbol{\Theta}} \mathcal{L}_R$ since it requires backpropagating through an entire simulation. The alternating scheme allows us to economically draw a large amount of historical snapshots, and to evaluate $\nabla_{\boldsymbol{\Theta}} \mathcal{L}_R$ only, for the autoencoder training.

**Continuous vs. One-Shot Autoencoder Training** Since robot dynamics change with changing control and design, continual retraining is critical. See, for example, Fig. 4 a. (robot arm control). With one-shot autoencoder training using only initial motion, the autoencoder only disambiguates motions of a mostly static arm, and optimization fails.

Our algorithm has no obvious guarantee of convergence; here, we describe three specific ways this algorithm can theoretically fail, and the steps we take to make our algorithm work reliably in practice.

**Overfitting to Historical Snapshots** It is important to fit to an entire trajectory, and not just the trajectory's individually captured historical snapshots. Overfitting the autoencoder to history will degrade feature quality on future scenarios. Therefore, we employ *early stopping* to be conservative with autoencoder training. Before training the autoencoder, we evenly split the snapshots into a training and validation set. We early stop the training when the validation loss has remained worse than the best seen loss value, for a certain number of consecutive iterations.

**Overfitting to Recent Trajectories** The autoencoder tends to prioritize learning the most recent trajectories, harming generalization to future snapshots. To remedy this problem, we maintain an *experience replay buffer* [Mnih et al., 2015] of snapshots from multiple simulations. We use all snapshots currently in the replay buffer to train the autoencoder. This increases the diversity of autoencoder training inputs, and stabilizes our algorithm against a changing controller.

---

**Algorithm 1** Learning-In-The-Loop Co-Optimization

---

**Hyperparameters:** Max episode $K$, Max optimization iterations $M$, max learning iterations $N$, minibatch size $b$, max replay buffer size $B$, target update step size $\alpha$, latent space dimensionality $r$.
**Given:** user-specified robot morphology $\mathcal{R}$, loss function $L$, design parameter bounds $\phi_{\min}$, $\phi_{\max}$, initial design parameters $\phi_0$, and initial full state $\boldsymbol{v}^0$.

Randomly initialize network weights $\boldsymbol{\theta}$, and $\boldsymbol{\Theta}$ (with latent space of dimension $r$), and initialize autoencoder copy $\boldsymbol{\Theta}' \leftarrow \boldsymbol{\Theta}$.
Initialize empty replay buffer $\mathcal{I} \leftarrow [\,]$ with maximum size $B$.
**for** episode i = 1 ... K **do**
    **for** optimization iteration j = 1 ... M **do**
        Compute loss $l_j$ and simulation snapshots $\Upsilon_j$: $l_j$, $\Upsilon_j = \mathcal{S}(L, \boldsymbol{v}^0)$.
        Store snapshots $\Upsilon_j$ in $\mathcal{I}$.
        Update $\boldsymbol{\theta}$, $\boldsymbol{\phi}$ using the analytical simulation loss gradients $\nabla_{\boldsymbol{\theta}, \boldsymbol{\phi}} L$.
        Clamp physical design variables $\phi_i \leftarrow \max(\min(\phi_i, \phi_{\max}), \phi_{\min})$.
    **end for**
    Split $\mathcal{I}$ randomly and evenly into training set $\mathcal{I}_\tau$ and validation set $\mathcal{I}_v$.
    **for** learning iteration j = 1 ... N **do**
        **for** minibatch $1 \ldots \mathrm{len}(\mathcal{I}_\tau)/b$ **do**
            Sample minibatch $I_\tau$ from $\mathcal{I}_\tau$ (without replacement)
            Update $\boldsymbol{\Theta}'$ using analytical autoencoder loss gradients $\nabla_{\boldsymbol{\Theta}} \mathcal{L}(\Upsilon)|_{\Upsilon = I_\tau}$
        **end for**
        Compute validation loss $\ell_j = \mathcal{L}(\mathcal{I}_v)$
        **if** $\ell_j$ has not decreased for $q$ iterations **then**
            Break (Early Stopping).
        **end if**
    **end for**
    Update Autoencoder weights using target: $\boldsymbol{\Theta} \leftarrow \boldsymbol{\Theta}'\alpha + (1 - \alpha)\boldsymbol{\Theta}$
**end for**
**Return:** $\mathcal{R}$ with optimized design $\phi$ and controller $\boldsymbol{\theta}$.

---

**Feature Oscillation** Despite our precautions thus far, the autoencoder can still change rapidly, which injects instability to the controller optimization. Inspired by the smoothed target network update scheme in Lillicrap et al. [2015], we perform learning on a *copy* of the autoencoder network weights, and use the original autoencoder network throughout an episode. After each episode, we step the original autoencoder toward the updated copy $\boldsymbol{\Theta}'$; *i.e.*, $\boldsymbol{\Theta} \leftarrow \boldsymbol{\Theta}'\alpha + (1 - \alpha)\boldsymbol{\Theta}$.

We combine these refinements with our learning and optimization phases in our final algorithm (See Alg. 1 for details).

## 5 Results and Discussion

In this section, we summarize the results of our experiments on four of our model robots: 2D Biped, 2D Arm, 2D "Elephant," 2D "Bunny," and briefly describe demonstrations on four further robots: 2D Rhino, 3D Quadruped, 3D Curved Quadruped, and 3D Hexapod. Robot morphologies can be seen in Figs. 4 and 6. The Biped and its variants are co-design experiments, while the others are pure control, as we found co-design matters much more for locomotion tasks. We encourage the reader to watch the accompanying video for simulations of our optimized robots. For each 2D example, we compare to another automated procedure, a $k$-means clustering baseline, inspired as an automated version of the manual labels from [Hu et al., 2019]. In this baseline, the particles were clustered prior to optimization based on their Euclidean distance in the robot's rest configuration; the average position and velocity of each cluster in each Cartesian coordinate is fed as input to the controller network. Further details about our hyperparameters are included in the Appendix for reproducibility. Each experiment was run ten times. All results are presented with a $90\%$ confidence interval. We provide some ablation tests in each experiment to justify the necessity of certain aspects of our algorithm. For all 2D experiments, iteration vs. loss is presented; autoencoder training time is trivial compared to simulation, and both the VAE and $k$-means simulations/backpropagation times are similar. Each 2D simulation and corresponding backpropagation is computed in less than 20 seconds. All experiments were performed on a computer with an Intel i7 2.91-GHz processor, NVIDIA GeForce GTX 1080 GPU, and 16GB of RAM.

**2D Arm** The 2D arm presents the simplest of all of our tasks in which the centroid of a region of a fixed-base soft robot arm must reach a prescribed point in space with no gravity. While geometrically simple, the problem is not dynamically trivial. The actuators are too weak to allow the robot to directly bend to to the goal; it must swing back and forth to build up momentum to reach its target. This is the easiest problem we present; in this one example, $k$-means clustering is competitive, and at finer resolutions, faster converging than our VAE (though slower at coarser resolutions). See Fig. 4 a.

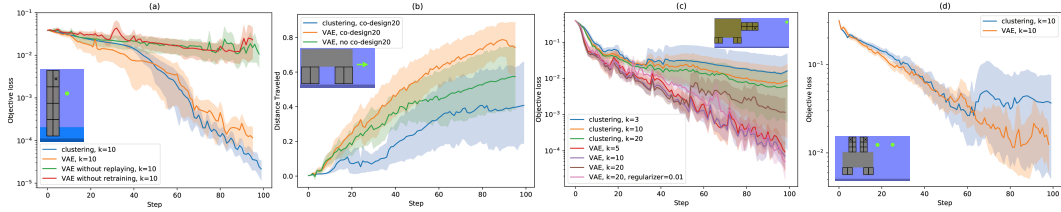

Figure 4: Progress of robot performance vs. optimization iteration, along with drawings of our 2D demos. Each black rectangle denotes an actuated region; in precision tasks, regions denoted with black $X$'s are those which must reach target locations, denoted with green circles.

We use the 2D arm as an opportunity to present ablation tests for what happens if the replay buffer is eliminated. In the former case, the representation oscillates wildly, making control optimization impossible. In the latter case, since earlier iterations have less dynamic motion, they provide less dynamically descriptive, insufficient representations of the robot's full motion.

**2D Biped** The 2D Biped presents a locomotion task in which the robot must run to the right as far as possible in the allotted time. The biped's progress can be seen in Fig. 4 b. In the video, we present two design variations in the robot's shape. We also show the results of a typical VAE training procedure in Fig. 5, showing typical convergence. In four out of the ten trials, $k-$means clustering completely failed to converge, landing in poor local minima near the robot's starting configuration.

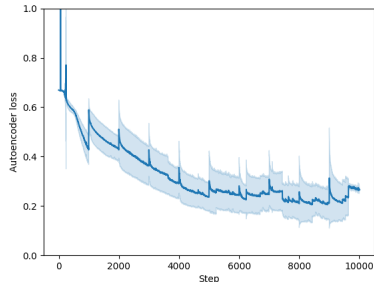

Figure 5: The average reconstruction loss $\mathcal{L}_R$ per pixel for the 2D Biped scaled as measured in average pixel distance vs. stochastic gradient descent step, demonstrating that not only does our algorithm converge in objective value, but also in model learning.

We further use the Biped as an opportunity to show the minimal adverse effects of retraining. Table 1 presents the change in the distance traveled after retraining the autoencoder and then performing a single optimization step. The single optimization step cancels out virtually all backward progress caused by retraining.

| Retrain # | Mean | Standard Dev. | Retrain # | Mean | Standard Dev. |
|---|---|---|---|---|---|
| 1 | $-1.08 \times 10^{-2}$ | $2.64 \times 10^{-2}$ | 9 | $-8.53 \times 10^{-3}$ | $6.07 \times 10^{-2}$ |
| 2 | $-1.68 \times 10^{-2}$ | $1.53 \times 10^{-2}$ | 10 | $-1.26 \times 10^{-2}$ | $1.80 \times 10^{-2}$ |
| 3 | $-1.25 \times 10^{-2}$ | $3.06 \times 10^{-2}$ | 11 | $-2.70 \times 10^{-5}$ | $2.61 \times 10^{-2}$ |
| 4 | $3.56 \times 10^{-2}$ | $8.36 \times 10^{-2}$ | 12 | $-1.00 \times 10^{-2}$ | $1.36 \times 10^{-2}$ |
| 5 | $-1.86 \times 10^{-2}$ | $2.88 \times 10^{-2}$ | 13 | $-6.54 \times 10^{-3}$ | $4.29 \times 10^{-3}$ |
| 6 | $-1.17 \times 10^{-2}$ | $1.28 \times 10^{-2}$ | 14 | $5.93 \times 10^{-3}$ | $1.16 \times 10^{-2}$ |
| 7 | $-4.13 \times 10^{-3}$ | $1.55 \times 10^{-2}$ | 15 | $-3.27 \times 10^{-4}$ | $8.39 \times 10^{-3}$ |
| 8 | $-9.69 \times 10^{-3}$ | $1.44 \times 10^{-2}$ | | | |

Table 1: The mean backward progress remaining from retraining after a single optimization iteration on the 2D Biped locomotion task, with corresponding standard deviations. A negative value indicates a decrease in the distance traversed. As can be seen, the backward progress is a very small negative number, or positive in all cases, indicating that a single optimization almost completely reverses the adverse effects of retraining.

**2D Elephant** The 2D Elephant presents a task which is a mixture of locomotion and manipulation. The elephant must walk to the right while a part of the trunk must reach a prescribed location. A subset of the results are seen in Fig. 4 c for readability, further results can be found in the Appendix. We use the Elephant to perform experiments over a wide range of latent variable and cluster sizes. While we try to compare the same number of clusters and latent variables in experiments (since inputs from a cluster give highly dependent data) we acknowledge that each cluster (in 2D) provides six inputs to the controller. Thus, this experiment also allows comparisons over controllers of similar size.

The VAE dominates $k$-means over all cluster/latent variable counts. As can be seen, the latent variable procedure has a weakness that the autoencoder can suffer the well-known "collapse" phenomenon as the latent variable size grows; increasing the VAE's regularizer weight combats this phenomenon.

**2D Bunny** The 2D Bunny provides a task in which two arms must reach two target locations in space. The robot must walk forward and bend the arms to reach the target points as closely as possible. This is our most dynamically challenging task and cannot be solved perfectly. Results are in Fig. 4 d.

**Further Demonstrations** Finally, we present further control tasks, including extensions to 3D and curvier designs (Fig. 6). Like the 2D Biped, these robots must run as far to the right as possible in the alotted time. The 2D rhino is instantiated directly from a .png file. 3D optimizations take much longer, since the 3D autoencoder, and the corresponding simulation data, are much larger. They take more time to process, and the VAE capacity is larger, meaning it requires larger minibatches, a larger replay buffer, and more iterations during learning. Each 3D walker takes over a day to optimize, and thus was only performed once; please see the video for demonstrations of the four additional walkers.

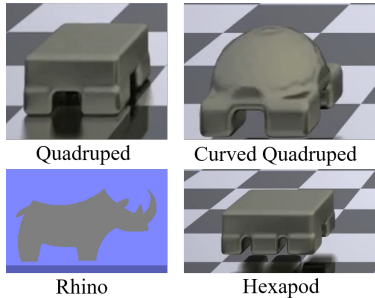

Quadruped      Curved Quadruped

Rhino      Hexapod

Figure 6: Four further robot demonstrations we present in more detail in the supplementary video.

**Discussion** As can be seen, our VAE observer tends to converge faster and get stuck in poor local minima much more rarely than $k-$means clustering. A natural question is to ask why $k-$means clustering performs worse. There are several reasons why clustering-based observors can lead to worse outcomes. First, $k-$means clustering can lead to poorly selected regions to track on the robot. For illustrative purposes, Fig. 7 shows two clusterings of the bunny. In the first, because the Euclidean distance is used, clustering leads to two segments that are geometrically close but *dynamically* dissimilar to be clustered together; especially a problem when the task demands that they ideally should be tracked separately (we note that a geodesic distance might not suffer as seriously from such a behavior). Second, clustering does not gracefully handle changes in robot feature size. Even though the top arms are more dynamically interesting than the body of the robot, the body is allocated the majority of the clusters. This can be compensated for in a brute-force manner by adding more clusters, but experiments showed that as the number of clusters grows large, simulation time slows tremendously (if, say, $k = 1000$ is used, simulation on the same problems can take minutes). Finally, clustering is dynamics-oblivious; it cannot adapt as different motions are explored, nor does it consider other design or task specifics like actuator placement.

Fig. 8 provides a visualization of the extracted latent features for the 2D Biped and describes their computation. The emergence of natural "physical modes" arises as the procedure continues, with some more significant latent feature representing more rigid motion modes (such as velocity to the right), and less significant latent features representing higher-frequency, dynamic deformations. Such representations are not only valuable for robust control, but can make it easier to understand learned observers and controllers in relation to the underlying physical processes.

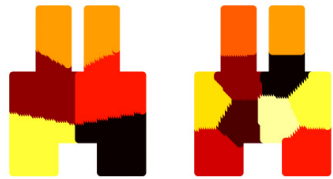

Figure 7: Two suboptimal clusterings for the bunny with different $k$ values. For $k = 5$ (left) the upper arms are clustered together; for $k = 10$ (right) clusters overemphasize the importance of the body compared with the feet or arms.

In order to understand how the mapping between velocity fields and actuations changes over time, we generated saliency maps for each frame of a 2D Elephant simulation, which is included in the supplemental video. The saliency maps show the gradient of each actuator control signal with respect to the x and y velocities, multiplied point-wise by the velocities. The gradients of the leg actuators (rows 1-4 from the top) with respect to latent variables are similar to one another, as is true for the trunk actuators (rows 5-10), implying similar parts of the task rely on similar latent coordinates.

Finally, we note that while our VAE observer dominates on more challenging problems, $k-$means is still sufficient for simpler problems, as can be seen with the arm. One other example where the $k-$means observer performs better is in the case of the 2D biped when the problem is made sufficiently easier, by giving it much stronger actuators and dropping it from a high initial height to give it initial momentum. In this case, the walker can quickly learn to "bounce" forward; however, in the much more difficult case shown here, the $k-$means observer often completely fails.

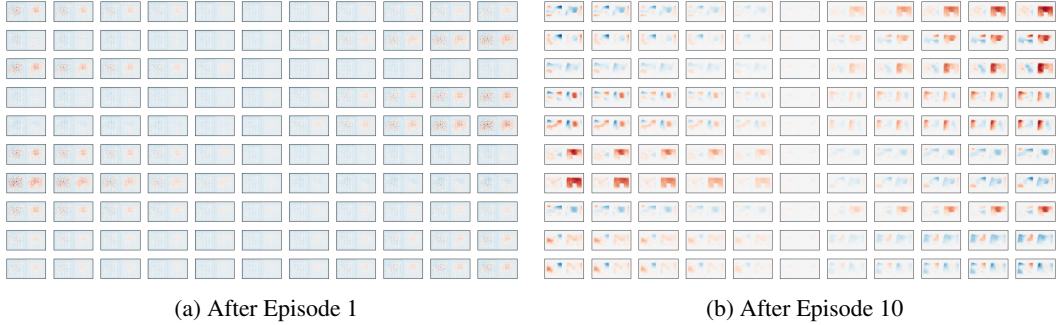

<div align="center">(a) After Episode 1                          (b) After Episode 10</div>

Figure 8: Visualization of the latent space of the autoencoder (2D Biped) with 10 latent variables. Each row represents a (normalized) decoded output for a one-hot latent feature, varied from $-1$ to 1. As the algorithm proceeds, the latent features become more descriptive. Formatting per box is the same as in Fig. 1.

# 6    Conclusions and Future Work

In this work, we demonstrated a method for end-to-end co-optimization of soft robots that requires minimal human intervention. Our method interleaves optimization with learning a deep latent space representation, allowing improved state estimates to improve the control and design, and vice versa. We have demonstrated our algorithm's superior reliability to naïve dynamics-oblivious methods.

Our method has two notable drawbacks. First, although the 2D version of our algorithm can be applied on a visual cross-section in 3D, the fully 3D version can be hard to realize in the physical world. Further, autoencoder training times can be quite large in the 3D convolutional architecture. Second, retraining of the autoencoder, while necessary, can sometimes undo some forward progress and interfere with momentum in optimization, both slowing the tail-end of convergence in optimization.

Finally, we envision three future extensions to our work. First, since we learn a low-dimensional latent space, it would be interesting to use the learned latent-space outside of the context of its counterpart controller - namely for, *e.g.* optimal control such as LQR control. Second, we would like to apply our control algorithm to other soft robot simulation methods. For example, the nodes of an FEM simulation could be similarly rasterized to a background grid (though with additional overhead; MPM generates this grid "for free") to which our algorithm could be directly applied. Finally, we hope to demonstrate our optimized controllers on real, physical soft robots using vision-based sensors and optical flow.

# 7    Acknowledgments

We thank Alexander Amini for insightful discussions on convolutional variational autoencoders and starter code. We thank Liane Makatura for help in drawing explanatory figures. We thank Buttercup Foshey (and of course Michael Foshey) for moral support during this work.

This work was supported by NSF grant No. 1138967, the Unity Global Graduate Fellowship, IARPA grant No. 2019-19020100001, and The MIT EECS David S. Y. Wong Fellowship.

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
