[Supplementary Material]

# Learning-In-The-Loop Optimization: Appendix

**Andrew Spielberg, Allan Zhao, Tao Du, Yuanming Hu, Daniela Rus, Wojciech Matusik**
CSAIL
Massachusetts Institute of Technology
Cambridge, MA 02139
aespielberg@csail.mit.edu, azhao@mit.edu, taodu@csail.mit.edu
yuanming@mit.edu, rus@csail.mit.edu, wojciech@csail.mit.edu

## 1    Problem Hyperparameters

The following hyperparameters are the same for all problems. Note that we work in a fictitious unit system for convenience.

| Hyperparameter | Value |
|---|---|
| Autoencoder learning rate | $1 \times 10^{-4}$ |
| Adam Parameter $\beta_1$ | 0.9 |
| Adam Parameter $\beta_2$ | 0.999 |
| Early Stopping Iterations $q$ | 10 |
| Minibatch Size | $\frac{1}{40}$ dataset size |
| Simulation $\Delta t$ | $5 \times 10^{-2}$s (with 100 substeps) |
| Simulation Time $T$ | $4\ s$ |
| Target Weight $\alpha$ | 0.75 |
| Poisson ratio | 0.3 |
| Density (kg / m$^3$) | 1.0 |
| Allowed Young's Modulus variability | $\pm 10\%$ |

Table 1: Fixed Hyperparameters

The following hyperparameters were used for each problem. The following rules were used to choose these hyperparameters. *1)* Controller learning rate is most dependent on the dynamics of the system, and can be the same for any representation, but the more sensitive the system is to actuation the lower the learning rate should be. *2)* The slower the objective changes, the larger $M$ should be. *3)* The larger the physical footprint of the robot and larger the dynamic range of its velocity profile, the more training iterations are necessary for convergence. *4)* The faster the trajectory of the system changes between episodes, the larger the replay buffer should be.

Young's Modulus is purely a parameter of the problem and was chosen to make the problem most challenging and interesting.

| Hyperparameter | 2D Arm | 2D Biped | 2D Elephant | 2D Bunny |
|---|---|---|---|---|
| Controller learning rate | $5 \times 10^{-4}$ | $5 \times 10^{-3}$ | $5 \times 10^{-3}$ | $5 \times 10^{-3}$ |
| Max Optimization Iter. $M$ | 6 | 6 | 10 | 10 |
| Max Learning Iter. | 10 | 15 | 20 | 30 |
| Replay Buffer Size | 4800 | 4800 | 2400 | 2400 |
| Young's Modulus (kg / (m s$^2$)) | 30 | 10 | 30 | 50 |

Table 2: Variable Hyperparameters

## 2 Complete 2D Elephant Experimental Results

Figure 1: Progress of robot performance vs. optimization iteration for the 2D Elephant example.