[Reviews · NeurIPS 2019]

Reviewer 1



Summary The paper proposes a differentiable pipeline that can jointly learn a latent space representation (via variational autoencoder) for controlling soft robots and optimize for the controller and the soft robot parameters for tasks in simulation, such as making a soft 2D robot walk forward as fast as possible. The work is made possible by using a differentiable hybrid-particle-grid based soft material physics simulator. The authors provided insightful details on the alternative minimization scheme for training the autoencoder, the controller neural network, and the robot parameters in tandem. The proposed framework was evaluated on 5 simulated experiments that show controllers using the learned representation outperforms ones using the baseline representation obtained via k-means clustering. Review While the performance of the system is impressive, the motivation of the approach is not well-communicated in 3 folds: In discussing the proposed hybrid-particle-grid based soft robot representation vs finite element methods, the authors claim that the high “degrees of freedom of finite element methods is impractical for most modern control algorithms.” While probably true, the authors provide no additional details to back up this statement, and it’s not clear why one can’t learn a latent representation over FEMs to act as the control input. The authors also claim that previous work using FEM simulation requires a priori knowledge of how robot will move as a drawback, but isn’t the hybrid-particle-grid simulator also “a prior knowledge”? The authors do not sufficiently explain why co-adapting a soft robot’s design parameters along with task and latent representation learning is desirable. Potential drawbacks of the approach include that learning the latent representations, controller, and robot parameters together, driven by loss of a single task, does not lead to representations and robot parameters that are useful for other tasks: The advantage of a fully differentiable pipeline over learning the representation and the controller is clear in the context of implementation and perhaps efficiency at learning a specific task. But, this does not mean that the learned latent representation is sufficient for tasks that it was not trained on, and it is also possible that representation is not generalizable to different downstream controllers. A similar point on task-generalization can be made about co-adapting the soft robot parameters in tandem with representation and controller learning. The optimized soft robot parameters (i.e. the young’s modulus of the robot materials) may have overfitted to the task and the controller since the only way that the robot’s fitness is determined is through the performance of a particular task. Of course the paper’s focus is on multi-task learning for soft robotics. This appears to weaken the argument for learning the representation explicitly over learning the controller directly for the task. Evaluating this alternative approach, which is simpler to train as it avoids the alternating training scheme, would strengthen the paper’s position. Additional comments: The specific task loss function was not provided in the paper. There is no discussion on convolutional autoencoder needs to scale w/ robot particle grid size While the authors state that they’ve chosen the particular autoencoder architecture based on “stability and generality across experiments,” they do no explained what stability and generality across experiments specifically mean, and it’s also not clear how (in)sensitive the architecture is to soft robots with varying complexities. The algorithm section contains many insights on how the authors made the alternating minimization scheme to work. In particular, training is done alternatively between the parameters of the controller + robot, and the latent representation. The authors implement conservative early stopping for the encoder to prevent overfitting to historical snapshots, experience replay buffer to prevent overfitting to recent trajectories, and target network to update and copy to source every once in awhile w/ a learning rate. There is concern however, that a training procedure with as many moving parts as this may be difficult to tune for new tasks and robots. The authors should also discuss other recent works on co-design and hardware representation approaches (just a couple listed): Wang et. al. Neural Graph Evolution, ICLR 2019 Chen et. al. Hardware Conditioned Policies for Multi-Robot Transfer Learning, NuerIPS 2018 Pathak et. al. Learning to Control Self-Assembling Morphologies: A Study of Generalization via Modularity, https://arxiv.org/abs/1902.05546 Conclusion: The proposed method is novel and the experiments are impressive, but because the motivation could be better articulated and method seems difficult to reproduce, the paper as it stands may not be as useful as its novelty. In conclusion a borderline accept is recommended.

Reviewer 2



This paper presents a hybrid algorithm (learning-in-the-loop optimization). This algorithm is tested on different robots in 2D and 3D. Although learning-in-the-loop optimization is not new in itself, the way the authors use it is interesting. The convolutional variational autoencoder is used to reduce the dimensionality of the robot, which is a very common control strategy. Instead of using the classic control theory, the surrogate observer seems a promising or good alternative way given the increasing power of modern computation. While close loop learning and optimization is promising, it does shows the problems such as stability and convergence. Another drawback of this method is it may works for robot that is with regular shape. The examples shown in the paper are all block-based robot which might not represent the real-world robot well. Also, representing the robot in 2D grid seems over-simplified. The methods relies on a fullly differentiable simulator, which may be another limitation as not many differentiable simulators are available.

Reviewer 3



This paper presents a method for end-to-end co-optimization of soft robots. The proposed algorithm is based on learning low-dimensional robot state while simultaneously optimizing robot control and/or material parameters. This is realized through a learning-in-the-loop co-optimization algorithm in which a latent state representation is learned as the robot solves a task. Originality: the proposed method is a novel interesting solution to controlling soft bodies in simulation; the paper frames the contribution within relevant related work, highlighting the differences of the proposed method to the state-of-the-art, including both classical control methods and data-driven approaches. Quality: the submission is technically sound and claims are supported by adequate analysis of the experimental results; authors also underline limitations and constraints of their methods (e.g. different performance with different working setups and design choices), which give valuable insight and understanding of the proposed method. Clarity: the paper is clearly written and organized, the description of the method is clear, and pseudo-code supports the text. It would be interesting to clarify the following points: - how realistic is it to assume a large dataset of robot motion data that is "*representative of the way the robot will move* when completing the prescribed task"? If part of the optimization goal is to actually learn how to perform the task, where is the initial dataset coming from? - how much did the feature oscillation affect the experiments? do you have quantitative results on the instability given by this effect? Significance: this is an interesting piece of work that can have a good impact in the field of soft robotics modeling and control in simulation. The experimental setup is simple but diverse enough to evaluate the proposed method. It would be interesting to see how the proposed method perform with more realistic models of soft bodies, possibly including multiple bodies interacting. The experiments presented seem however a good starting point for demonstrating the new proposed method.

[Author Response · NeurIPS 2019]

We thank the reviewers for their constructive feedback. While there are many papers on rigid robot control each year at
NeurIPS, control of soft robots has seldom been addressed. Our submission is novel, bridging learning and soft robotics,
and is the first to tackle end-to-end control and co-optimization of soft robots of arbitrary morphology. Our algorithm
takes natural advantage of fully differentiable simulation, which is exploding in popularity and relevance[1,2,3].

We appreciate the reviewers' compliments that our submission is *"an interesting piece of work that can have a good*
*impact in the field of soft robotics..." (R3)*, *"very novel learning work on soft robots"* with *"impressive"* experiments
and performance *(R1)*, and that the approach is *"promising" (R2, R3)* with *"potential to be relevant for future work..."*
*(R3)*. We believe concerns can be addressed within the review cycle with text improvements and additional experiments.

**More Complex, Non-Blocky/3D Designs (R2, R3).** We can easily add more complex, non-blocky and 3D designs to
the results and video. MPM particles are flexible enough to represent most irregular geometry found in soft robotics and
CNNs can adequately learn over such inputs. We include a few new results below. We hope these new results complete
a convincing gamut of experiments already described as *"impressive," (R1)* and *"diverse"* and *"promising" (R3)*.

**a)** A new rhino robot loaded from image, serving as a curvy, non-blocky 2D example. **b)** Convergence of the rhino control task over 10 trials. The topheavy, unactuated head makes this a challenging control task. **c)** With 24 actuators and highly nonlinear dynamics, this 3D Hexapod is now our most complex demo. After 100 optimization iters., it runs 1.5 body lengths in $4s$. **d)** This new 3D Quadruped's hemispherical body proves our method works on less blocky 3D shapes as well. After 100 optimization iterations, it runs two body lengths in $4s$.

**Clarification on FEM vs. MPM for Learning (R1).** We are not saying that one cannot learn a latent representation
on FEM nodes; in fact, it is possible our approach could extend to FEM by rasterizing node velocities to a grid and
directly applying our method. However, such an approach has never been demonstrated. We chose MPM because *1)*
prior work[1] demonstrates its success for control, as it naturally handles differentiable contact, and *2)* it acts directly on
a velocity grid, providing a representation amenable to CNNs for free.

**Why a Latent Space Is Necessary (R1).** It is indeed impractical to try to learn directly on FEM node or MPM particle
coordinates. This approach doesn't scale: we tried feeding 1000 MPM particles into our controller (a relatively small
number), and runtime for simulation and backprop ballooned by $10\times$ compared to compact latent features. As a further
advantage, velocity grids can easily be captured in the real world *via* optical flow; dense node coordinates cannot.

**Simulation as Prior Knowledge (R1).** We do not consider the *simulator*, be it FEM or MPM, as prior knowledge, but
rather the *data* it generates. In previous work the robot is simulated along many random trajectories to build a prior
dataset. If the dynamics of the target trajectory are not explored initially, the observer and resulting optimization suffer.
This issue is especially salient during design optimization, where system dynamics change. LITL continually generates
representative data throughout the optimization phase and re-learns, thus it does not suffer this drawback.

**Initial Dataset Generation (R3).** A small initial dataset is generated from simulating just once with the initial,
untrained controller. This is enough to bootstrap our learning.

**Benefits of Co-Optimization and Co-Learning (R1).** The value of co-optimization has been highlighted in prior work
for rigid[4,5] and soft robots[1]; it allows robots to solve difficult tasks more easily and improve performance. We can
add comparisons of performances with and without co-optimization. The latent space must be co-learned since the
experienced dynamics (and thus, optimal observer) change during co-optimization.

**Problem Scope (R1).** R1 wrote *"of course the paper's focus is on multi-task learning for soft robotics."* We wish to
clarify that this paper's focus is *not* on multi-task learning, but single task learning with highly dynamic soft robots. As
*R3* states, *"this is a challenging problem."* It is one that has seldom been successfully tackled by any literature; ours is
the first end-to-end co-design solution for general morphologies. While multi-task learning would be an interesting
extension, it is yet another very hard problem, one rich enough for its own dedicated manuscript.

**Training Stability (R1, R2, R3).** R2 and R3 asked how feature oscillation affects *"stability and convergence."* "Back-
ward progress" from feature oscillation is dominated by the following optimization phase and typically undone within
1-2 iterations. We can add tables quantifying this effect. *R1* asked why we couldn't learn *"the representation explicitly*
*over learning the controller directly for the task."* We tried a joint optimization, (see section 4.3, **Alternative vs.**
**Simultaneous Minimization**), but the CNN layers learned too slowly to provide useful signal, leading to bad local
minima. Alternating minimization avoids this issue in economical fashion.

[1] Hu, Yuanming, et al. "ChainQueen: A Real-Time Differentiable Physical Simulator for Soft Robotics." 2019 IEEE International Conference on Robotics and Automation (ICRA). IEEE, 2019.
[2] de Avila Belbute-Peres, Filipe, et al. "End-to-end differentiable physics for learning and control." Advances in Neural Information Processing Systems. 2018.
[3] Schenck, Connor, and Dieter Fox. "SPNets: Differentiable Fluid Dynamics for Deep Neural Networks." Conference on Robot Learning. 2018.
[4] Ha, Sehoon, et al. "Joint Optimization of Robot Design and Motion Parameters using the Implicit Function Theorem." Robotics: Science and systems. 2017.
[5] Spielberg, Andrew, et al. "Functional co-optimization of articulated robots." 2017 IEEE International Conference on Robotics and Automation (ICRA). IEEE, 2017.


[Meta-Review · NeurIPS 2019]

This is a borderline paper showing how to learn a latent representation using a VAE for control of soft robots. There were concerns by the reviewers about how the results could be generalized and scaled to other simulations and systems. However, there was consensus that the work was novel and should be presented at NeurIPS.